# Drag Reduction Technology of Water Flow on Microstructured Surfaces: A Novel Perspective from Vortex Distributions and Densities

**DOI:** 10.3390/ma16051838

**Published:** 2023-02-23

**Authors:** Chunye Liu, Wene Wang, Xiaotao Hu, Fulai Liu

**Affiliations:** 1Key Laboratory of Agricultural Soil and Water Engineering in Arid and Semiarid Areas, Ministry of Education, Northwest A&F University, Yangling 712100, China; 2Department of Plant and Environmental Sciences, University of Copenhagen, 1353 Copenhagen, Denmark

**Keywords:** drag reduction, riblet surface, superhydrophobic surface, vortex density, water flow structures, Ω_M_ vortex identification method

## Abstract

Revealing the turbulent drag reduction mechanism of water flow on microstructured surfaces is beneficial to controlling and using this technology to reduce turbulence losses and save energy during water transportation. Two microstructured samples, including a superhydrophobic and a riblet surface, were fabricated near which the water flow velocity, and the Reynolds shear stress and vortex distribution were investigated using a particle image velocimetry. The dimensionless velocity was introduced to simplify the Ω vortex method. The definition of vortex density in water flow was proposed to quantify the distribution of different strength vortices. Results showed that the velocity of the superhydrophobic surface (SHS) was higher compared with the riblet surface (RS), while the Reynolds shear stress was small. The vortices on microstructured surfaces were weakened within 0.2 times that of water depth when identified by the improved Ω_M_ method. Meanwhile, the vortex density of weak vortices on microstructured surfaces increased, while the vortex density of strong vortices decreased, proving that the reduction mechanism of turbulence resistance on microstructured surfaces was to suppress the development of vortices. When the Reynolds number ranged from 85,900 to 137,440, the drag reduction impact of the superhydrophobic surface was the best, and the drag reduction rate was 9.48%. The reduction mechanism of turbulence resistance on microstructured surfaces was revealed from a novel perspective of vortex distributions and densities. Research on the structure of water flow near the microstructured surface can promote the drag reduction application in the water field.

## 1. Introduction

Long-distance flow transportation is required due to the uneven distribution of water resources. However, the energy loss of water flow is huge during transportation [1,2]. Channels and pipelines that require pumping energy are the applications that need surface drag reductions to save energy and resources [3]. In 2020, there were over 10 million kilometers of water supply pipelines in China [4,5]. Therefore, it is urgent to find an energy-saving way to solve the surface drag problem of water flow. Various microstructured surfaces, constructed by micro- or nano-structures, are proposed to reduce the surface drag and energy loss [6]. Microstructured surfaces, originating from the imitation of the rough surfaces from plants and animals in bionics, are an effective means to reduce the surface drag. For example, Dewdrops can roll freely on lotus leaves due to the existence of rough structures on the surface, while the riblet skin of sharks receives less resistance when moving [7]. Superhydrophobic surface is a kind of microstructured surface, with high hydrophobicity, which is extremely difficult to be wetted by water. The contact angle of water droplets on superhydrophobic surfaces is greater than 150°. Constructing different microstructured surfaces, reducing the dissipation of energy and the surface frictional resistance when the water makes contact with surfaces, is considered as one of the reliable methods to improve drag reduction rates.

Methods of constructing microstructured surfaces mainly include phase separation, electrochemistry, sol–gel, spraying, and etching [8,9,10]. Among those methods, the spraying and etching methods are relatively simple for fabricating microstructured surfaces. The spray method is mainly employed to construct a rough structure with low surface energy by spraying, while the etching method is to process microstructures by mechanical processing, laser, etc. Those two fabricating methods do not rely on expensive equipment or complicated operating procedures.

Many types of microstructured surfaces have been studied, from superhydrophobic surfaces with slip properties to those such as Sharkskin riblet surfaces [11]. Microstructured surfaces have been widely used in antifouling, microfluidics, and drag reduction [12,13,14]. In particular, the reduction mechanism of turbulence resistance on microstructured surfaces mainly includes two aspects. The drag reduction effects of microstructured surfaces, including different shapes and sizes of the surface, have been extensively studied by experimental methods [15,16]. Most people thought that the shape of the blade’s drag reduction effect was the best. However, this structure was unstable and difficult to exist on the surface for a long time. Therefore, it has been proposed to use V-shaped riblets to reduce drag. The V-shaped riblets had good drag reduction performance when the dimensionless spacing *s*^+^ of riblets was between 10 and 20 [3]. Some scholars assessed that when *s*^+^ of the V-shaped riblet was 15, the reduction effect of drag was the best [17]. On the other hand, the drag reduction mechanism of the microstructured surfaces was analyzed using numerical simulation. The drag reduction impact of superhydrophobic surfaces was studied by the numerical simulation method, proving that the slippage of the water flow along the streamwise reduced the frictional resistance. The shear stress-free boundary of the superhydrophobic surface is the main factor that changes the flow velocity near the surface through the pressure drop experiment of the microchannel [18]. Gas and still water filled in the riblet valley contribute to reducing the effective contact area, decreasing the velocity of the boundary layer, and saving water energy.

Changes in vortex structures of water flow near surfaces are affected by the microstructures when water flow is transported. Therefore, the identification of vortex structures on surfaces is crucial. Firstly, in the identification method of vortices, it was believed that the curl of the velocity vector, that is, the vorticity, could represent the vortex. However, this cognition was immature. Since 1980, the identification method of vortices had been produced, namely, the *Q*, λ_2_, and Δ methods, which can vaguely represent the strength of the vortex, and the result of those methods was a scalar [19]. Then, new identification methods of vortices, namely, the Ω, Liutex vector, objective Ω, and Liutex-Ω method, etc., overcoming the shortcomings of previous identification methods of vortices, have emerged [20,21,22]. Vortex structures of water flows on microstructured surfaces were different compared with smooth surfaces [23,24]. Therefore, applying vortex identification to microstructured surfaces can provide a further understanding of the drag reduction mechanism.

The construction method, shape, size, and drag reduction effects of microstructured surfaces are mainly carried out in the existing research on turbulent drag reduction. As water flow fields near microstructured surfaces are not easy to observe, there are few studies on the influence of structure surfaces on water flow resistances. Existing studies have shown that microstructured surfaces changed the vortex structure of water flows [25]. The development and evolution of vortex structures have become an important starting point for the drag reduction study on microstructured surfaces [26,27]. With the development of the particle image velocimetry technology and the supporting hardware equipment, the improved performance provides a test platform for observing the flow field on microstructured surfaces.

In this paper, the Ω method is improved to simplify the identification of vortex structure. A new created vortex identification method Ω_M_ is used to identify vortex structures. Referring to the concept of atmospheric vortex density in meteorology, a novel definition of vortex density in water flow is proposed to quantify the vortex density. The mechanism of drag reduction on microstructured surfaces is analyzed from the perspective of the vortex distribution position and the vortex density, which provided a theoretical basis and technical support for the use of microstructured surfaces. Studying the water flow structures on microstructured surfaces can give more support for the low-drag application of the surfaces in water flow, so as to realize energy saving and environmental friendliness.

## 2. Experimental Devices

### 2.1. Experimental Materials

A superhydrophobic and a micro-riblet surface were fabricated using acrylic plates by the spraying method and the etching method, respectively; their structures are shown in Figure 1. A smooth acrylic plate served as a control. The superhydrophobic surface was sprayed on the acrylic plate with never-wet superhydrophobic spray (Rust-oleum, Vernon Hills, IL, USA), composed of micro–nano-scale structures. To increase the durability of the coating, the surface of the plates was roughened with sandpapers and then washed with water and alcohol, respectively. The distance between the spray and the surface was kept the same, and the method of spraying horizontally first and then vertically was adopted to ensure the same spray thickness while making uniform spraying. Structures of superhydrophobic surfaces were observed by scanning electron microscope (Hitachi, Tokyo, Japan). Superhydrophobic surfaces were sampled at several different locations. It was found that the shapes and sizes of the microstructures at different positions were similar to the papillary structures on the surface of lotus leaves, as shown in Figure 1a. The contact angle of superhydrophobic surfaces was 150.9°, measured by JY-PHB contact angle meter measurement with the accuracy of 0.1° (Jinhe Instrument Manufacturing Co., Ltd., Chengde, China). It should be noted that these represent a conservative estimate of the contact angle of the superhydrophobic surface, because the experiment used the water droplets with a volume of 2 μL. When 5 μL of water drops was used, the water drops fell directly from the contact surface. This is why the contact angles measured by our test were smaller than that provided by the manufacturer (160–170°). The V-shaped riblet surfaces were fabricated with a V600 CNC machine tool (Dahe CNC Machine Co., Ltd., Yinchuan, China; the positioning accuracy is ±0.005/300 mm) at a depth *h*_1_ of 0.8 mm and an angle *α* of 90° [15,17], as shown in Figure 1b.

### 2.2. Experimental Design

A low-velocity water circulation system was designed for experimentation, which included a circulation open channel and a two-dimensional PIV system, working simultaneously in Figure 2. The circulation open channel system was composed of a water storage tank, a rectangular water channel with a shrinking section at the head, a water pump, a water stabilization tank (a layer of circular hole-shaped water stabilization network was arranged in the middle of the stabilization water tank), two valves, a tailwater tank, and a water delivery pipe. The channel was 1200 mm in the streamwise direction (*x*), 150 mm high in the normal direction (*y*), and 120 mm wide in the spanwise direction (*z*), which was made of transparent acrylic and equipped with two layers of water stabilization nets at the shrinking section. Flow velocity was adjusted by the tailgate at the end of the channel to form a uniform flow. At the bottom of the channel, an acrylic plate was laid. The test position was embedded with 100 mm-long microstructured plates, located 900 mm from the channel inlet and 200 mm from the tailgate (Figure 2). Meanwhile, the riblet plate was placed in the streamwise direction. A cylindrical tripwire with a diameter of 4 mm was arranged at the inlet of the channel to allow the boundary layer to transition to a fully developed turbulent. A small-scale circulating water channel matching with microstructured surface can be used to represent the water channel in practical applications, mainly because the results of our paper are all dimensionless.

The PIV system used in the experiments (Cube World Co., Ltd., Beijing, China) was mainly composed of four parts: the image (SM-CCDB5M16), double-pulse laser (Vlite-200), synchronization controller, and data analysis system. The parameters of the laser are as follows: the laser wavelength, energy, and thickness are 532 nm, 200 mJ, and 1 mm, respectively. The maximum shooting frequency was 15 Hz, while the pulse width was ≤8 ns. The particle image resolution was 2456 pixels × 2058 pixels. Tracer particles were hollow glass microbeads with a density close to that of water. The main component of the hollow glass microbeads was SiO_2_ (content is greater than 65%), and the average particle size was 10 μm. The smaller particle size ensured that the particles had a good flow following property. It was necessary to ensure enough tracer particles in the near-surface area during the test. When the number of tracer particles near the surface was small, the tracer particles shall be added appropriately.

In particle image processing, the cross-correlation calculation and iterative algorithm were used concurrently, and the initial interpretation area was 32 pixels × 32 pixels. Based on the of the previous calculation, the size of the interpretation area was reduced, and the size of the interpretation area changed from 32 pixels to 16 pixels, and then 8 pixels. The iterative algorithm was used to improve the signal-to-noise ratio of cross-correlation calculation and the accuracy of calculation results. At the same time, an image bias algorithm was introduced, and the windows overlap was 50%. The minimum size of the grid in the interpretation area was 0.31 mm × 0.31 mm. The Gaussian fitting method was used to obtain the sub-pixel calculation error (that is, the accuracy of the calculation result is ±0.1 pixels). Meanwhile, the minimum cross frame time for shooting was 2000 μs to 3000 μs. The view field of the captured image was 90 mm × 70 mm (flow direction × normal direction), and the actual resolution of the captured area was 0.037 × 0.036 mm/pixel.

Experiments were carried out to observe the flow fields of the three surfaces, namely, the superhydrophobic surface (SHS), riblet surface (RS), and smooth surface (SS), at different Reynolds numbers. The smooth surface was used for control. The Reynolds number Re (Reynolds numbers were characterized by the length of the microstructured plate from the head of the large plate, Re = *u*L/*ν*, *u* is the average flow velocity, L is the length of the microstructured plate from the head of the large plate, L = 900 mm, *ν* is the kinematic viscosity) ranged from 80,173 to 148,893. In the experimental Reynolds number range, the dimensionless spacing *s*^+^ (*s*^+^ = *su**/*ν*, *s* is the riblet spacing, *u** is the friction velocity) of the V-shaped riblets was 10–20, which had a good drag reduction effect [3]. Three treatments with Reynolds numbers of 85,900 (Re1), 120,260 (Re2), and 137,440 (Re3) were taken as examples for detailed analysis. Experimental treatments are shown in Table 1.

Experimental plates were placed horizontally at the bottom of the channel, and the laser was adjusted so that the laser position was perpendicular to the camera shooting position; their locations are illustrated in Figure 2. Clear particle images were captured by adjusting the laser intensity and camera focal length, and 700 images were recorded for each treatment. The average velocities of the measured section were obtained by time averaging. When the laser passes through the junction of the water flow and the solid surface, a reflected noise signal was generated. To prevent the noise signal from affecting the experimental results, the black background paper was pasted on the back and bottom of the test section.

### 2.3. Statistical Analyses

Micro-Vec software V3.6.1 in the PIV system was used to obtain tracer particle images. Seven hundred images were captured in each group of treatments. The cross-correlation calculation method was used to calculate the two frames before and after the shooting to obtain the instantaneous velocity distribution in the flow field. Based on the transient flow velocity, the average flow velocity distribution, Reynolds shear stress, drag reduction rate, and vortex structures near the surface region can be calculated. The calculation formulas are as follows.

The average velocity distribution of the logarithmic layer is [28]
(1)$$u^{+}=\frac{1}{\kappa }\ln y^{+}+B$$
where *u*^+^ is the dimensionless velocity, and *y*^+^ is the normal dimensionless position, specifically:(2)$$u^{+}=\frac{u}{u^{*}},\;y^{+}=\frac{yu^{*}}{\nu }$$
where *κ* is the Karman constant; *u* is the average flow velocity, m/s; B is a constant number; *y* is the normal distance, m; and *ν* is the kinematic viscosity of the water flow, m^2^/s.
(3)$$u=\frac{1}{\kappa }u^{*}\ln y+u^{*}(\frac{1}{\kappa }\ln\frac{u^{{}_{*}}}{\nu }+B)$$

The wall friction velocity *u*^*^ can be calculated by logarithmic fitting with the velocity (*y*, *u*) of the logarithmic law layer. The calculation formula of wall friction shear stress *τ* is [29]
(4)$$\tau =\rho u^{*2}$$

The dimensionless Reynolds shear stress $\tau_{xy}^{+}$ is calculated by the Formula (5):(5)$$\tau_{xy}^{+}=\frac{-\rho u^{\prime }v^{\prime }}{u_{\infty }^{2}}$$
where *ρ* represents the water density; *u′*, *v′* are the fluctuation velocity in the streamwise and normal direction, respectively; and *u*_∞_ is the free flow velocity.

The drag reduction rate *DR* is
(6)$$DR=\frac{\tau -\tau_{0}}{\tau_{0}}\times 100\%$$
where *τ* is the frictional resistance of the microstructured surface, kg/ms^−2^, and *τ*_0_ is the frictional resistance of the smooth surface, kg/ms^−2^.

## 3. Modification of Ω Method

Vortices of the turbulent flow field near the surface can be changed by microstructures. At present, the new vortex identification method—Ω method—which has the characteristics of capturing vortices of different intensity, is not sensitive to the threshold [20]. The threshold value of the Ω method is 0.52, which means vortices exist when the calculated value of the Ω method is greater than 0.52. The Ω method decomposes the velocity gradient into two parts. A rotating part is represented by an antisymmetric tensor B. The non-rotating part is represented by a symmetric tensor A, which represents pure shear. This method is the ratio of the rotating vorticity to the total vorticity. The formula is
(7)$$\Omega =\frac{\|B{\|}_{F}^{2}}{\|A{\|}_{F}^{2}+\;\|B{\|}_{F}^{2}+\varepsilon }$$
where‖‖*_F_* is the Frobenius two-norm of the matrix. In practical applications, a small positive parameter *ε* is added to the denominator to prevent the denominator from being zero when the sum is zero. *A*, *B* stands for a symmetric tensor and an antisymmetric tensor. Existing studies have used empirical values [30].
(8)$$\varepsilon =0.002(\|B{\|}_{F}^{2}-\|A{\|}_{F}^{2})=0{.001Q}_{\max}$$
where *Q*_max_ is the maximum value of the *Q* vortex identification method.

When using the Ω method to identify vortices, it is necessary to first calculate the *Q*_max_ using the *Q* method, making the application process of the Ω method complicated. In this paper, the maximum values *Q*_max_ calculated by the *Q* method were analyzed. At the same Reynolds number, the vortices of different plates change. However, the *Q*_max_ value would not change significantly, as shown in Figure 3. Therefore, using dimensionless velocity instead of *Q*_max_ is a choice of simplified vortex identification method. After introducing the dimensionless velocity, the correction ε is
(9)ε=0.001U
where U represents the dimensionless flow velocity, which is the ratio of the average flow velocity (unit: m/s) to the unit velocity, and the coefficient 0.001 is to ensure that ε is a small positive number [30]. The improvement of this parameter solves the problem that the denominator is 0 while simplifying the Ω method.

The calculation formula of the Ω_M_ method using the dimensionless flow velocities is established, and the calculation formula is:(10)$$\Omega =\frac{\|B{\|}_{F}^{2}}{\|A{\|}_{F}^{2}+\;\|B{\|}_{F}^{2}+0.001U}$$

A large number of vortices at different strengths exist in turbulence. The number of vortices at different strengths and positions is not the same, which makes the characteristics of energy transfer and consumption different. In meteorology, the vortex density W (the total number of vortices per unit volume or area) is used to describe the distribution of vortices in the atmosphere [31]. The vortex density W_Ω_, introduced into the water flow, is the ratio of the total number of a strength vortex to the total number of all strength vortices at the unit volume or area. The relative vortex density W_Ω_ is:(11)$$W_{\Omega }=\frac{S_{\Omega }}{S}\times 100\%$$
where *S*_Ω_ is the total number of a strength vortex determined by the Ω_M_ method, and *S* is the total number of all strength vortices. Compared with the total number of all strength vortices, the relative vortex density W_Ω_ can be used to clarify the influence of the proportion of vortices at different strengths on the transfer and consumption of energy in the turbulent flow.

## 4. Results and Discussion

### 4.1. Average Velocity Distribution

The averaged velocity of the boundary layer was obtained based on the instantaneous velocity fields obtained by the PIV system. The dimensionless time-averaged velocity *u*^+^ and the normal dimensionless distance *y*^+^ were calculated according to Formula (7). *U^*^* was obtained by nonlinear iterative fitting according to Formula (8). Figure 4 shows the distribution of *u*^+^ and *y*^+^ on microstructured surfaces. The velocity *u*^+^ increased with the increase in *y*^+^, presenting an evident growth trend, and velocities of the boundary layer had an apparent partition phenomenon in different regions of *y*^+^. The turbulent boundary layer was composed of a viscous sublayer (0 < y^+^ < 5), a buffer layer (5 < y^+^ < 30), and a logarithmic layer (30 < y^+^ < 300). Compared with the SS, the average flow velocity of microstructured surfaces began to increase on the buffer layer. From the enlarged view of the logarithmic, the velocity distribution of the logarithmic layer, from large to small, was the SHS, RS, and SS, respectively. The flow velocity distribution of the two microstructured surfaces was obviously shifted outward, showing that the average flow velocity of microstructured surfaces was greater than that of the smooth surface at the same normal position, and microstructured surfaces had a specific drag reduction effect. The velocity shift to the outer layer represented the drag-reducing properties of microstructured surfaces [32,33]. Analyzing the reduction mechanism of turbulence resistance on microstructured surfaces, it was found that microstructures of the SHS had a gas–liquid interface, and the water droplet and the surface were prone to relative slippage [34]. Daniello et al. thought that the superhydrophobic surface had the characteristics of micro–nano rough structures and low surface energy, which reduced the frictional resistance of the surface and increased the velocity of the boundary layer [18]. From the scanning electron microscope images of the SHS at 500 μm and 1 μm (Figure 1), it can be seen that micron-scale protrusions with irregular shapes, which include nano-scale fine structures, were distributed on the surface, making it easier for water flow to slip along the solid surface at the intersecting interface. Meanwhile, the low energy of the SHS made it difficult for water droplets to adsorb on the surface. Lee et al. maintained that the average velocity curve of the riblet surface moved up due to the reason that the velocity in the riblet valley was relatively small, resulting in stable low-velocity streaks [35]. Meanwhile, the tip of the riblet broke flow direction vortices and generated a large number of secondary vortices. The generation of secondary vortices weakened the turbulence of water flows and reduced the downward sweep of high-speed water flow. Changes in these flow structures reduced surface velocity.

### 4.2. Reynolds Shear Stress

Reynolds shear stress is the shear stress induced by the intermixing of the fluctuation velocity in turbulent flow, which represents the exchange of unit fluid momentum per unit area [36]. In the investigation of the characteristics of drag reduction on microstructured surfaces, the variation in the Reynolds shear stress can effectively reflect the pulsation variation in the water flow.

Figure 5 shows the Reynolds shear stress change in the boundary layer between microstructured surfaces and the SS with the normal distance *y*^+^ at different Reynolds numbers. At the buffer layer, the Reynolds shear stress increased with the increase in *y*^+^. The Reynolds shear stress of microstructured surfaces was decreased compared with the SS, which presented that the surface can restrain the turbulence of water flow and reduce the fluctuating velocity. The Reynolds shear stress on the buffer layer, from large to small, was the SS, RS, and SHS. The reduction in the Reynolds shear stress on microstructured surfaces was due to the fact that microstructured surfaces attenuated the fluctuating flow velocity, thereby reducing the Reynolds shear stress [37]. Due to the existence of the riblet structures, the flow velocity of the riblet surface was small, and stable low-speed streaks were generated. The existence of riblet structures suppressed the possibility for the burst of low-velocity streaks, resulting in a reduction in the fluctuating velocity of water flow, which was ultimately manifested as a reduction in the Reynolds shear stress [38]. From the energy point, the riblet surface restricted the development of the fluctuating flow velocity, indicating a decrease in the momentum exchange in water flow and leading to the drag reduction [17]. When located at the logarithmic layer, the Reynolds shear stress of the two types of microstructured surfaces decreased. However, the Reynolds shear stress of three different surfaces no longer showed significant differences.

Figure 5 shows the change in the Reynolds shear stress at different surfaces. The maximum Reynolds shear stress occurred at the junction of the buffer layer and the logarithmic layer. Table 2 shows the percentage R of the maximum Reynolds shear stress reduction in microstructured surfaces compared with the SS at each working condition. The R values of the SHS and RS ranged from −9.8% to −15.9% and −2.9% to −3.5%, respectively. The reduction in the Reynolds shear stress indicated that the fluctuating flow velocity decreased, and microstructured surfaces had a certain inhibitory effect on the water flow turbulence. At the same Reynolds number, the Reynolds shear stress of the superhydrophobic surface had the most significant reduction, indicating that it had the best suppression effect on flow turbulence.

### 4.3. Vortex Distribution

Studying the distribution of vortices on microstructured surfaces is helpful in understanding the drag reduction process. Water flow generates moments with the action of shear stress, leading to the rotation of the water flow and thus generating vortices. Vortices, the essential feature of the turbulent water flow, are the medium of energy transfer and dissipation.

By studying the variation in vorticity near the surface of microstructures, it is found that vorticity values increase as they get closer to the surface. Figure 6 shows the maximum value of vorticity near surfaces. When the Reynolds number was 85,900, the maximum value of the SHS and RS was 26 s^−1^ and 29 s^−1^, respectively. The vorticity value of the SS is 34 s^−1^. Compared with the SS, the vorticity of microstructured surfaces was significantly reduced near the surface, indicating that microstructured surfaces had a drag reduction effect. Distribution of the near-surface vorticity gradually decreased with the increase in the surface distance, and the highest vorticity value was on the near-surface region.

#### 4.3.1. Reliability Analysis of the Ω_M_ Method

According to the division method of the flow area, the boundary layer was in the range of 0 < y < 0.3 h, and h stood for the water depth [39,40]. In this paper, the water flow area within 0.3 h was first analyzed, as illustrated in Figure 7. Figure 7a,b show the distribution of vortices on the near-surface region by the Ω and the Ω_M_ method when the Reynolds number is 85,900, respectively. The vortex distribution obtained by the two methods was the same. However, the ability of the Ω_M_ method to identify strong vortices was enhanced, as indicated by the marked positions in Figure 7. The distribution of vortices determined by the Ω_M_ method was reliable, and the method did not need to obtain the maximum value of the *Q* method, which simplified the model and retained the advantages of the Ω method.

#### 4.3.2. Vortex Structures on Drag Reduction Effects

The drag reduction mechanism of microstructured surfaces was analyzed from the two aspects of vortex distribution and vortex density. Figure 7 shows the vortex distribution near the surface of microstructures. The streamwise and normal coordinates in Figure 7 were dimensionless, where the abscissa was the ratio of the streamwise coordinate x to the lengths of shooting area b (x/b), and the ordinate was the ratio of the normal coordinate y to the water depth h (y/h). Analyzing the vortex distributions determined by the Ω_M_ method (Figure 7b–d), the SHS and the RS had a significant decrease in vortices within 0.1 h compared with the SS. The weak vortices were far away from the surface, proving that microstructured surfaces had a specific inhibitory effect on the burst of vortices. Within 0.2 h, the distribution of strong vortices on microstructured surfaces was significantly weakened, which is affected by the interaction of vortices within 0.1 h, showing that microstructured surfaces are inhibited within 0.2 h. This result was consistent with Nezu [39]. However, the strong vortex of microstructured surfaces no longer showed a significant weakening trend compared with the smooth surface within 0.2 h–0.3 h, which meant microstructured surfaces were difficult to affect the vortex change in this range. It can be seen from Figure 7b that vortices greater than 0.7 were less on the microstructured surface than on the SS, indicating that the microstructured surface inhibited the development of strong vortices. This result was consistent with Martin [41]. Compared with the riblet surface, the SHS had relatively fewer vortices greater than 0.7. Therefore, the SHS had a stronger inhibitory effect on strong vortices. In addition, vortices greater than 0.52 on the SHS had a more obvious tendency to move away from the surface [34]. Vortices on the SHS were significantly reduced within 0.1 h. Similarly, Figure 7c,d can also provide the same conclusion.

Vortex density at different levels near the surface was analyzed. Vortex density in Figure 7 was divided into five levels, which were 0.52–0.6 (the 5th level), 0.6–0.7 (the 4th level), 0.7–0.8 (the 3rd level), 0.8–0.9 (the 2nd level), and 0.9–1 (the 1st level). The vortex values of 5th level were the smallest, and the vortex values of 1st level vortices were the largest. The vortex density was introduced to quantify the different level vortices, as shown in Figure 8. The Ω_M_ method had an enhanced ability to identify strong vortices compared with the Ω method.

Comparing (a) and (b) in Figure 8, the 1st level vortex density identified by the Ω and Ω_M_ method remained the same. The difference between the 2nd and 3rd level vortex density was small. However, the 4th and 5th level vortex density identified by the Ω_M_ method decreased. The 4th and 5th level vortex density accounted for more than 75%, playing a dominant role in the turbulence of the water flow near the surface. Adrian also proved that weak vortices played a major role near the surface [25]. Meanwhile, the vortex density of the 5th level vortex was the largest, and the minimum value of the 5th level vortex density in all working conditions was 49.6%, accounting for about half of all vortices. Figure 8b shows the vortex density of the three surfaces (SHS, RS, and SS). The 5th level vortex density of the microstructured surface increased (65.5% and 63.9% for the SHS and RS, respectively) compared with the SS (59.0%) at the same Reynolds number. The weak vortices near the surface increased, which showed that the microstructures had a significant inhibitory impact on vortices. These results were the same as Zhang et al. [42]. Compared with the RS, the 5th level vortex density on the SHS was greater (65.5% > 63.9%), with the result being that the SHS had a stronger inhibitory effect on vortices and a better drag reduction effect [3,18]. The same conclusion can be drawn for other working conditions. The vortex density of the 4th level vortices on microstructured surfaces was reduced compared with that of the SS. However, the vortex density of the 4th level on the riblet surface was greater than that of the SHS and the RS. As shown in Figure 8b, the 4th level vortex density of the SHS and the RS was 25.6% and 27.1%, respectively. This may be due to the different drag reduction mechanisms of the two surfaces.

Comparing (b), (c), and (d) in Figure 8, the vortex density of the 5th level vortices decreased with the increase in the Reynolds numbers when the surface was the same. When the Reynolds numbers were Re1, Re2, and Re3 (Re1 < Re2 < Re3), the 5th level vortex density on the SHS was the largest, which decreased with the increase in the Reynolds number. The value of the 5th level vortex density on the SHS ranged from 54.8% to 65.5%, This showed that the SHS could inhibit the 5th level vortices within this Reynolds number range. At these three Reynolds numbers, the vortex densities of the 5th level vortices on the riblet surface were 63.9%, 53.5%, and 53.9%, respectively. The 5th level vortex density of the riblet surface no longer showed a decreasing trend when the Reynolds number increased to Re3, demonstrating that the drag reduction effect was weakened at this Reynolds number.

In general, the vortex density of the 5th level vortices on the two microstructured surfaces was greater than that on the SS, and the vortex density of the 1st level and 2nd level vortices decreased. The microstructured surface inhibited the development of the vortices, thereby reducing the frictional resistance [43]. Weak vortices were beneficial to the dissipation of turbulence, and they could reduce the sharp change in surface shear stress caused by the sudden instability of the strip-like structures, which was consistent with the previous analysis results of flow velocity and the Reynolds shear stress. The reason why the SHS had a better drag reduction impact was that the surface reduced the frictional resistance by reducing the interaction between the viscous force of the water flow and the surface, thereby weakening the water turbulence and increasing weak vortices [44]. The instability of water flow near the surface caused the burst of low-velocity streaks and the jet of vortices into the outer layer, resulting in the instability of streaks structures [45,46]. The burst of streaks and vortices was the fundamental factor leading to the increased surface frictional resistance [47]. The RS reduced the turbulence of the water flow by increasing the stability of the water flow in the riblet valley and caused an increase in the vortex density of weak vortices. Therefore, changing the surface structures to reduce the burst of water flow was beneficial in lowering the surface frictional resistance.

## 5. Drag Reduction Rate

It was found that both microstructured structures had a particular drag reduction effect within a Reynolds number range. The drag reduction rates of microstructured surfaces are shown in Table 3. The maximum drag reduction rates of the SHS and RS were 9.48% and 4.93%, respectively. By summarizing the existing research results, it was found that the drag reduction rate of microstructured surfaces studied by the numerical simulation research method was relatively large, while the experimental method showed that the results were smaller than that of the simulation. In this paper, the research was carried out using experiments, so the previous experimental research results were used as a reference. The drag reduction rate of the superhydrophobic layer ranged from 10% to 30% [34,48]. When the dimensionless spacing *s*^+^ was 10–20, the drag reduction rate of the 90° V-shaped riblet surface was about 5% [15,17]. The main reasons for the differences were that the fabrication technology of microstructured surfaces was different, which was the main factor affecting the final drag reduction rate.

## 6. Conclusions

The improved Ω_M_ method was established. The drag reduction mechanism of microstructured surfaces from the new perspective of the vortex distribution and the vortex density provided theoretical support for the application of microstructured surfaces. The velocity was introduced into the Ω method to simplify the method. The water flow velocity of microstructured surfaces increased, while the Reynolds shear stress decreased compared with the SS. The vortex density of the weak vortices on microstructured surfaces was greater than that of the SS, and the vortex density of the strong vortices decreased within 0.2 h. Compared with the RS, the SHS had a better drag reduction impact from the perspective of the vortex distribution and the vortex density, and the drag reduction rate was 9.48%. Microstructured surfaces with matching dimensions for channels or pipes can achieve surface drag reduction for water transportation. It is of great significance to study the turbulent drag reduction mechanism of water flow on microstructured surfaces using the improved Ω_M_ vortex identification method.

## Figures and Tables

**Figure 1 materials-16-01838-f001:**
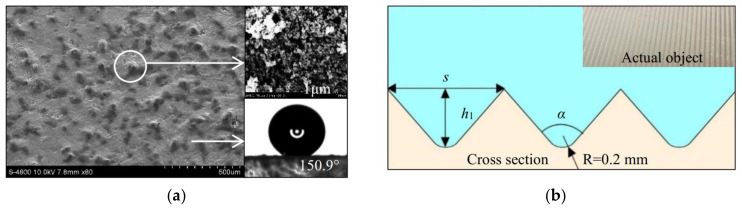
Schematic diagram of microstructured surfaces. (**a**) Electron microscope scanning image of superhydrophobic surface and surface droplet morphology. (**b**) Cross-section of the surface of V-shaped riblets and the actual object.

**Figure 2 materials-16-01838-f002:**
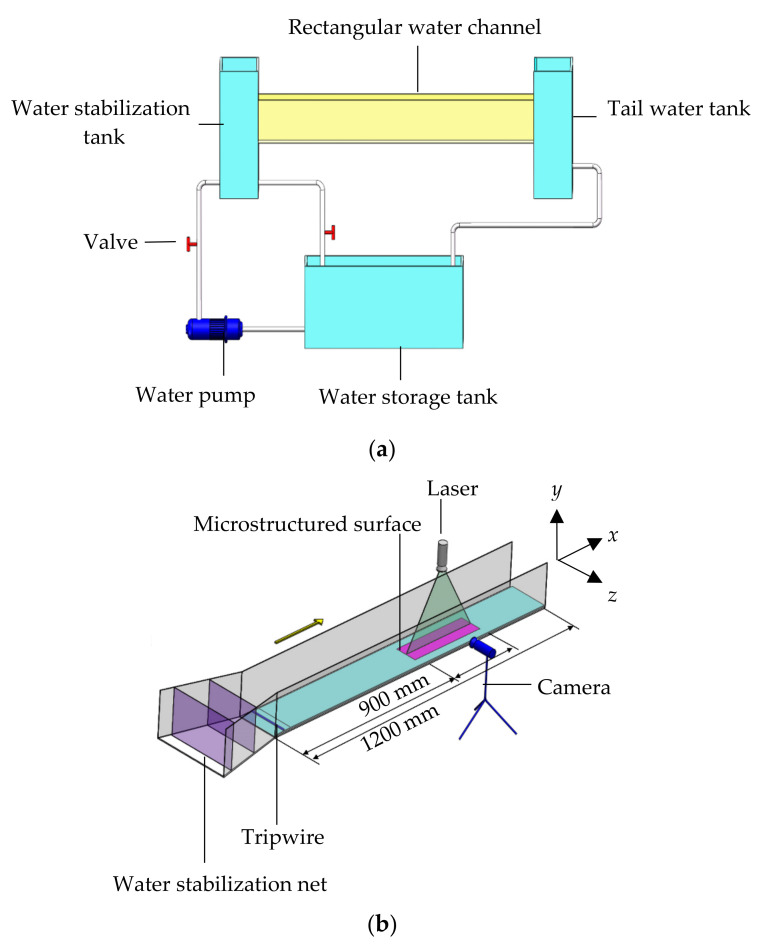
Schematic diagram of experimental devices. (**a**) Schematic diagram of experimental devices. (**b**) Rectangular water channel with a shrinking section.

**Figure 3 materials-16-01838-f003:**
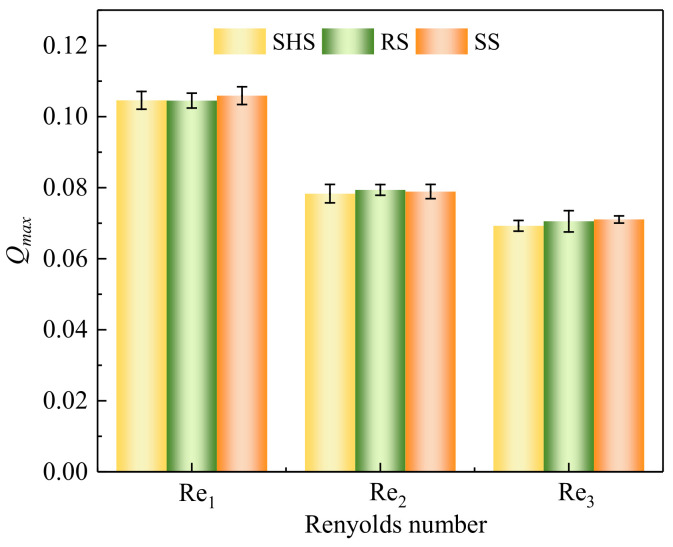
*Q*_max_ at different Reynolds numbers. Note: SHS, RS, and SS are the superhydrophobic surface, riblet surface, and smooth surface, respectively.

**Figure 4 materials-16-01838-f004:**
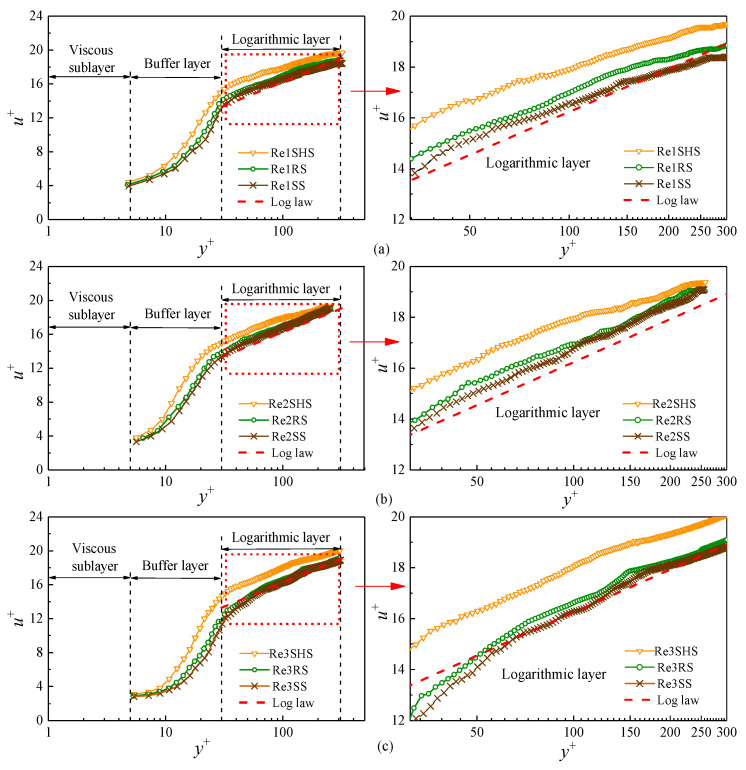
Dimensionless velocity *u*^+^ of microstructured surfaces along the normal distance *y***^+^**. (**a**) The velocity distribution of Re1. (**b**)The velocity distribution of Re2. (**c**) The velocity distribution of Re3.

**Figure 5 materials-16-01838-f005:**
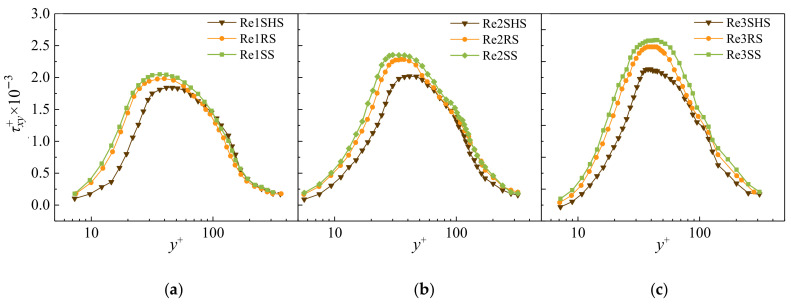
Variation in Reynolds shear stress $\tau_{xy}^{+}$ along the normal distance *y***^+^**. (**a**) Reynolds shear stress of Re1. (**b**) Reynolds shear stress of Re2. (**c**) Reynolds shear stress of Re3.

**Figure 6 materials-16-01838-f006:**
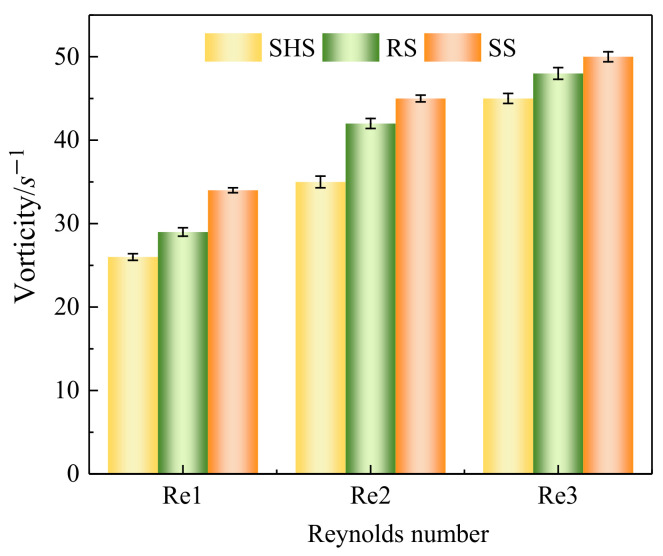
Maximum value of vorticity. Note: SHS, RS, and SS are the superhydrophobic surface, the riblet surface, and the smooth surface, respectively.

**Figure 7 materials-16-01838-f007:**
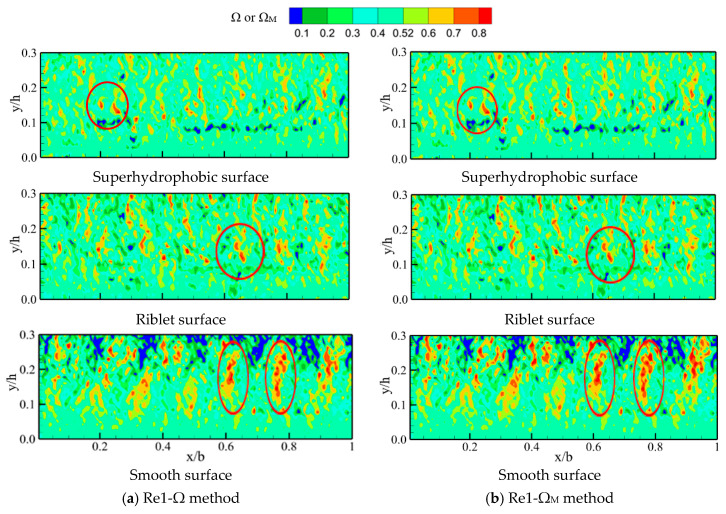
Vortex distribution near different surfaces at different Reynolds numbers. Note: Re1, Re2, and Re3 are the Reynolds numbers of 85,900, 120,260, and 137,440, respectively.

**Figure 8 materials-16-01838-f008:**
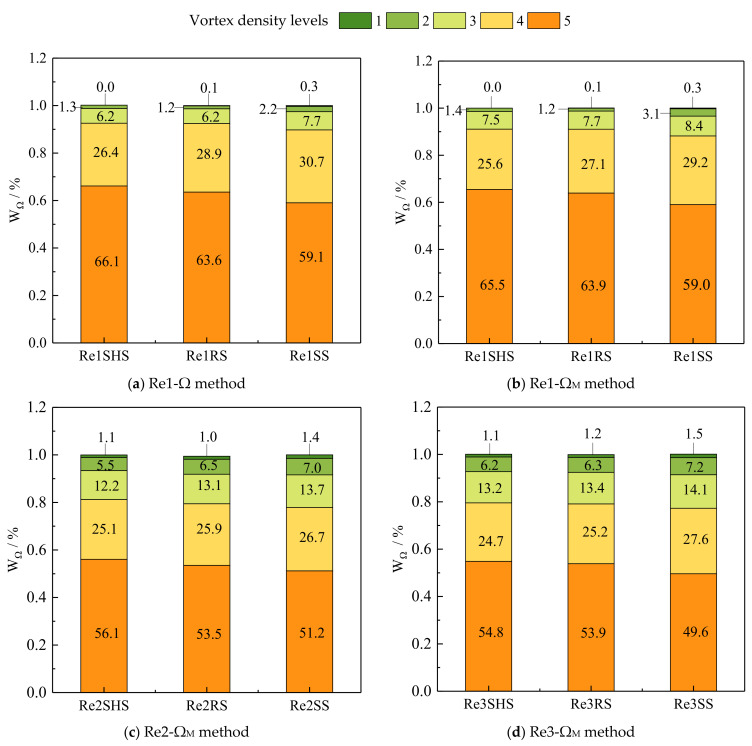
Vortex density near microstructured surfaces at different Reynolds numbers. Note: Re1, Re2, and Re3 are the Reynolds numbers of 85,900, 120,260, and 137,440, respectively. SHS, RS, and SS are the superhydrophobic surface, the riblet surface, and the smooth surface, respectively.

**Table 1 materials-16-01838-t001:** Experimental treatments.

Experimental Treatments	Reynolds Number	Experimental Surface	Experimental Treatments	Reynolds Number	Experimental Surface	Experimental Treatments	Reynolds Number	Experimental Surface
Re1SHS	85,900	SHS	Re2SHS	120,260	SHS	Re3SHS	137,440	SHS
Re1RS	85,900	RS	Re2RS	120,260	RS	Re3RS	137,440	RS
Re1SS	85,900	SS	Re2SS	120,260	SS	Re3SS	137,440	SS

**Table 2 materials-16-01838-t002:** Reduction in the maximum Reynolds shear stress.

Experimental Treatments	$\mathrm{Maximum}\;\mathrm{Value}\;\mathrm{of}\;\tau_{xy}^{+}/10{-}^{3}$	R/%	Experimental Treatments	$\mathrm{Maximum}\;\mathrm{Value}\;\mathrm{of}\;\tau_{xy}^{+}/10{-}^{3}$	R/%	Experimental Treatments	$\mathrm{Maximum}\;\mathrm{Value}\;\mathrm{of}\;\tau_{xy}^{+}/10{-}^{3}$	R/%
Re1SHS	1.84	−9.8	Re2SHS	2.02	−14.0	Re3SHS	2.38	−15.9
Re1RS	1.98	−2.9	Re2RS	2.28	−3.1	Re3RS	2.73	−3.5
Re1SS	2.04	0	Re2SS	2.35×	0	Re2SS	2.83	0

**Table 3 materials-16-01838-t003:** Drag reduction rate of microstructured surfaces at different experimental conditions.

Treatments	Friction Shear Stress *τ* (kg/ms^−2^)	Drag Reduction Rate (%)
Re1SHS	0.025	9.48
Re1RS	0.027	4.93
Re1SS	0.028	0.00
Re2SHS	0.034	8.58
Re2RS	0.036	4.50
Re2SS	0.037	0.00
Re3SHS	0.053	7.80
Re3RS	0.055	4.35
Re3SS	0.058	0.00

## Data Availability

The data presented in this study are available on request from the corresponding author wangwene@nwsuaf.edu.cn.

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
