# Peer review of "Drag Reduction Technology of Water Flow on Microstructured Surfaces: A Novel Perspective from Vortex Distributions and Densities"

_materials, 2023, doi:10.3390/ma16051838_

Round 1

Reviewer 1 Report

The present manuscript deals with the study of the influence of microstructure surfaces on the drag reduction technology of water flow. The authors made an experimental study using two-dimensional PIV. The obtained results are interesting. I recommend Minor revision. The authors should respond to the following comments:
1/ In Fig. 2, please replace pump by water.

2/ The diagram of the Fig. 2 is not clear, the authors should correct it. Name the two pictures as Fig. 2.a and Fig.2.b.
3/ Line 24, did the treatment or the acquisition take 10 mn? Please precise.
4/ In sections 2.3 and 3, I strongly recommend to add more details about the adopted methodologies. Please add some references.

Author Response

  1. In Fig. 2, please replace pump by water.

Response 1: Thank you for your suggestion. Fig. 2 has been modified. See Fig. 2 for details.

  1. The diagram of the Fig. 2 is not clear, the authors should correct it. Name the two pictures as Fig. 2.a and Fig.2.b.

Response 2: Thank you for your suggestion. Fig. 2 divided into Fig. 2(a), Fig. 2(b). See Fig. 2 for details.

  1.  Line 24, did the treatment or the acquisition take 10 mn? Please precise.

Response 3: Thank you for your suggestion. The meaning here is not that all experiments take 10 minutes, but it takes 10 minutes to process 700 pictures. To avoid ambiguity, we deleted 10 minutes. See line 222 for details.

  1. In sections 2.3 and 3, I strongly recommend to add more details about the adopted methodologies. Please add some references.

Response 4: Thank you for your suggestion. References has been added in sections 2.3 and 3. See line 228, 237 for details.

Reviewer 2 Report

Thank you for your highy quality study and paper. I suggest acceptance of the paper after the following minor revisions:

Point 1. The text needs to be corrected for grammar errors and corrections. Examples of these errors are;

  1. On line 47, the word “method” at the end of the sentence is unnecessary.   
  2. On line 46  "include the phase..."  "the" needs to delete. 

Point 2. The shifts in figure 2 texts need to be corrected.

Author Response

  1. The text needs to be corrected for grammar errors and corrections. Examples of these errors are;

On line 47, the word “method” at the end of the sentence is unnecessary. 

On line 46  "include the phase..."  "the" needs to delete.

Response 1: Thank you for your suggestion. The word “method” at the end of the sentence have been deleted. "The" have been deleted. See lines 62, 63 for details.

  1. The shifts in figure 2 texts need to be corrected.

Response 2: Thank you for your suggestion. Fig. 2 has been corrected. See Fig. 2 for details.

Reviewer 3 Report

Reviewer comments:

Manuscript Title: Drag reduction technology of water flow on microstructure sur- 2

faces: A novel perspective from vortex distributions and densities

Submission Date: Tuesday, Feb 07, 2023.

This paper is about two microstructure samples, including the superhydrophobic and riblet surfaces, were fabricated, near which the water flow velocity, Reynolds shear stress, and vortex distribution were investigated using a particle image velocimetry. The paper is interesting. It is suitable for publication after few changes. I recommend a major revision.

Address these comments:

(1)     [QUESTION] How can author improve the experimental values?

(2)     [SUGGESTION] Define the novelty of your analysis and provide references to the equations (if possible).

(3)     [QUESTION] Is it possible to improve the drag reduction rate 9.48%?

(4)     [LIMITATIONS] What are the limitations of the current study?

(5)     [QUESTION] The Reynolds number is high because of turbulence but Why this particular range is mentioned 85900 to 137440?

(6)     [SUGGESTION] Write few lines on vortex method.

(7)     [QUESTION]  What is superhydrophobic surface exactly?

The rest of my review presents other weak points, comments, and opinions in detail.
Overall Comments:

(1)     [KEYWORDS] The keywords (i.e., index terms) should be sorted in alphabetical order.

(2)     [ABSTRACT] The abstract should contain the best-achieved results from the performed analysis.

(3)    [SYMBOLS] The authors should add a table of symbols in the revised manuscript.

(4)    [CONCLUSIONS] The conclusions in this manuscript are primitive. Please, write your conclusions.

(5)    [PROOFING] The manuscript should be checked again to fix any typos such as missing spaces and commas.

(6)    [REFERENCES IN INTRODUCTION] Recent citations should be added to the manuscript.

Cite all of these studies:

DOI: 10.1007/s40430-019-1993-3,  DOI 10.1088/1873-7005/ab67d9

Author Response

(1) [QUESTION] How can author improve the experimental values?

Response 1: Thank you for your question. To reveal the turbulent drag reduction mechanism of water flow on microstructure surfaces. We use the Ω vortex method to identify the vortices in the flow. Meanwhile, a simplified ΩM vortex method is created. The newly created vortex identification method is beneficial to the application of drag reduction technology of water flow on microstructure surfaces. The value of this paper is to analyze the drag reduction mechanism of water flow on microstructure surfaces from a novel perspective.

(2) [SUGGESTION] Define the novelty of your analysis and provide references to the equations (if possible).

Response 2: Thank you for your suggestion. According to your suggestion, we emphasize the novelty of the paper in the introduction. See lines 113 to 116 for details. References of equations have been added. See the equation section for details.

(3) [QUESTION] Is it possible to improve the drag reduction rate 9.48%?

Response 3: Thank you for your question. The drag reduction rate 9.48% can be improved in other experiments. In this experiment, the focus of the research is to explore the drag reduction mechanism of microstructure surfaces through new methods. The value of drag reduction rate is only to show that the designed surface does have drag reduction effect. Only by understanding the drag reduction mechanism can we design the surface that can improve the drag reduction rate.

(4) [LIMITATIONS] What are the limitations of the current study?

Response 4: Thank you for your question. The limitation of current vortex identification methods is that the identification methods are relatively complex. How to use simple and effective methods to identify the vortex structure in water flows is the content that needs to be explored. Therefore, a simplified identification method of flow vortices is created in this paper.

(5) [QUESTION] The Reynolds number is high because of turbulence but Why this particular range is mentioned 85900 to 137440?

Response 5: Thank you for your question. In the experimental Reynolds number range, the dimensionless spacing s+ of the V-shaped riblets is 10-20, which has a good drag reduction effect. See lines 204 to 206 for details.

(6) [SUGGESTION] Write few lines on vortex method.

Response 6: Thank you for your suggestion. The vortex method is introduced in in the introduction part. See lines 89-101 for details.

(7) [QUESTION] What is superhydrophobic surface exactly?

Response 7: Thank you for your suggestion. The definition of superhydrophobic surface has been added in the introduction. See lines 56, 57, 58 for details.

The rest of my review presents other weak points, comments, and opinions in detail. Overall Comments:

(1) [KEYWORDS] The keywords (i.e., index terms) should be sorted in alphabetical order.

Response 1: Thank you for your suggestion. The keywords have been sorted in alphabetical order. See lines 27, 28 for details.

(2) [ABSTRACT] The abstract should contain the best-achieved results from the performed analysis.

Response 2: Thank you for your suggestion. Yes, the abstract contained the best-achieved results from the performed analysis. See lines 21, 22, 23 for details.

(3) [SYMBOLS] The authors should add a table of symbols in the revised manuscript.

Response 3: Thank you for your suggestion. The symbol table has been added. See the table before the introduction for details.

(4) [CONCLUSIONS] The conclusions in this manuscript are primitive. Please, write your conclusions.

Response 4: Thank you for your suggestion. The conclusions have been modified. See conclusions for details.

(5) [PROOFING] The manuscript should be checked again to fix any typos such as missing spaces and commas.

Response 5: Thank you for your suggestion. The manuscript has been checked again. See the red section for details.

(6) [REFERENCES IN INTRODUCTION] Recent citations should be added to the manuscript.

Response 6: Thank you for your suggestion. Recent citations have been added to the manuscript. See citations for details.
